# The relationship between physical activity and sleep quality among college students: The chain-mediating effects of self-control and mobile phone addiction

**Zhonggen Yin**[1,2], **Chengbo Yang**[1]\*, **Tong Liu**[1], **Jialiang Yu**[1], **Xiaomei Yu**[1,2], **Shuting Huang**[3], **Yanhong Zhang**[4]

**1** Institute of Sport Training, Chengdu Sport University, Chengdu, Sichuan, China, **2** College of Physical Education and Health Management, Chongqing University of Education, Nanan, Chongqing, China, **3** Chongqing Yucai Secondary School, Jiulongpo, Chongqing, China, **4** Sports Department, Southwest University of Political Science & Law, Shapingba, Chongqing, China

\* yangchengbo2024@sina.com

**Data Availability Statement:** All relevant data are within the manuscript and its Supporting Information files.

## Abstract

The psychological and physiological health of undergraduates was correlated with the sleep quality, which can be improved through increasing physical activity. However, the correlations between physical activity and sleep quality are subject to various factors. In this study, we investigated the effects of self-control and mobile phone addiction on the correlations between physical activity on undergraduates' sleep quality at the psychological and behavioral levels. Data was collected through a survey with a convenient sample of 2,274 students in China. The study utilized scales of physical activity, sleep quality, self-control, and mobile phone addiction to quantitatively evaluate the impact of physical activity on the sleep quality of undergraduates. The correlations were analyzed using SPSS 26.0, including descriptive statistics, confidence tests, common method bias tests, correlation analysis, and hypothesis tests. Pearson correlation analysis shows that physical activity was significantly correlated with sleep quality ($r = -0.541$, $p < 0.001$), and that physical activity and sleep quality were significantly correlated with self-control and mobile phone addiction. Regression analysis shows that physical activity had a significant positive regression effect on self-control (standardized regression coefficient $\beta = 0.234$, $p < 0.001$), a significant negative regression effect on mobile phone addiction ($\beta = -0.286$, $p < 0.001$), and a significant negative regression effect on sleep quality ($\beta = -0.351$, $p < 0.001$). Further, a chain mediation model of physical activity $\rightarrow$ self-control $\rightarrow$ mobile phone addiction $\rightarrow$ sleep quality was proposed. The findings provide basic data for college students to promote physical activity and improve sleep quality.

## 1. Introduction

Sleep is an important aspect of human life and is essential for maintaining one's physical and mental health [1]. However, sleep problems are prevalent throughout the world [2]. For

**Funding:** Yanhong Zhang received the award. The full name of funder: The Scientific and Technological Research Program of Chongqing Municipal Education Commission (Grant No. KJZD-K202200306).URL of funder website:https://jw.cq.gov.cn/.Zhang Yanhong participated in the data collection and analysis of the paper.

**Competing interests:** The authors have declared that no competing interests exist.

example, in the United States, 500,000 ~ 700,000 adults suffer from sleep disorders [3]. In China, the overall prevalence of insomnia is 17 percent [4]. The overall prevalence of poor sleepers in the Korean general population was 41.0%, with males accounting for 35.6% and females for 46.2% [5].The findings from the German Health Interview and Examination Survey for Adults reveal that approximately one-third of participants reported experiencing significant clinical issues with either initiating or maintaining sleep. Additionally, nearly one-fifth of respondents indicated that they frequently suffer from poor sleep quality [6]. Notably, the quality of sleep in the college population has been one of the major concerns of scholars [7, 8]. The sleep quality of university students is generally poor, rendering them susceptible to issues such as sleep deprivation, sleep disorders and sleep irregularities. Poor sleep quality may lead to chronic diseases such as obesity [9] and cardiovascular disease [10], and it may also trigger depression [9], mood disorders [11], and other health problems. These problems not only affect their studies and personal lives, but also jeopardize their physical and mental health [12]. In some provinces of China, the proportion of undergraduates with poor sleep quality has surpassed 20.0%; for example, the proportions in Guizhou, Hong Kong and Jiangshu were 53.7%, 57.5% and 26.64%, respectively [13–15].

Numerous studies have shown that physical activity can have a beneficial effect on sleep quality [16, 17]. In a study of adolescents, a moderate amount of physical activity was observed to be associated with better sleep quality [18]. A randomised controlled trial (RCT) reported that moderate-intensity physical activity was associated with a significant reduction in insomnia [19]. A cross-sectional research study conducted in Japan suggests that aerobic and relaxation exercise, along with reduced screen time, are important for improving subjective sleep quality in university students [20]. Overall, physical activity exhibited promising outcomes for improving sleep quality across diverse populations.

Although a number of randomised controlled trials and cross-sectional survey studies have examined the effects of physical activity on sleep quality, the underlying mechanisms of this process need to be further explored. The meta-analysis showed an association between lack of physical activity or sedentary behavior and an elevated risk of depression [21], which may also lead to the development of metabolic syndrome [22]. Additionally, prolonged LED screen use may significantly suppress melatonin, which in turn affects the biological clock and leads to sleep problems [23]. With the ubiquity of electronic devices in contemporary society, the influence of mobile phone use on the sleep quality of college students has become a significant concern. It is posited that enhancing individual self-regulation can mitigate the risk of mobile phone addiction [24, 25]. Furthermore, engaging in regular physical activity has been shown to bolster self-regulation [26, 27]. This raises the question of whether the positive effect of physical activity on sleep quality is mediated by a sequential process involving the enhancement of self-regulation and the reduction of mobile phone addiction. This hypothesis warrants empirical investigation as a promising research direction. Therefore, we undertook a cross-sectional study focusing on university students in China to delve into the mechanisms that connect physical activity and sleep quality among young individuals. The findings of our study contribute to a broader understanding of the interplay between physical activity and sleep quality. Moreover, the insights derived from this research may extend beyond the immediate context, offering valuable guidance for the development of interventions aimed at enhancing sleep quality and fostering physical and mental well-being. While our study was centered on Chinese university students, the identified relationships and potential interventions could be relevant and adaptable for college students in other geographic locations, provided that cultural and environmental factors are taken into consideration in the translation of these findings to different settings.

## 1.1 Correlations between physical activity and sleep quality

Sleep quality is an important factor in maintaining health and improving well-being [28]. Sleep quality has been defined as an individual's subjective perception of sleep [29]. The study by Che et al. found that good sleep quality is strongly positively associated with cardiovascular health, metabolic health, immune system health, and cognitive function, including reduced risk of cardiovascular disease, lower incidence of metabolic syndrome [30], etc. College students who engage in physical activity more frequently report lower anxiety levels and greater well-being [31], and those who spend more time on such activities exhibit a higher self-concept. Unfortunately, poor sleep quality has been a common phenomenon among college students, which manifested by inadequate sleep duration, difficulty falling asleep, daytime dysfunction, and persistent sleepiness. Although poor sleep quality can lead to lower academic performance and mental health issues for these students. According to a survey conducted by a mid-Atlantic university in the United States, 69 percent of college students are classified as poor sleepers [7]. A cross-sectional study found that only 48.7% of German university students reached the Pittsburgh Sleep Quality Index(PSQI) threshold for poor sleep quality [32]. The prevalence of poor sleep quality among Chinese college students was 39.42% [33].

Physical activity has been demonstrated to positively impact sleep quality. A systematic review study has shown that moderate physical activity benefits sleep quality in healthy people of all age groups [34]. A recent systematic review study also concluded that regular physical activity can improve sleep quality, reduce sleep latency, and enhance sleep efficiency [35]. Although the relationship between physical intensity and sleep quality lacks experimental evidence, the review demonstrates that moderate physical exercise is beneficial for sleep quality across all age groups within the healthy population [34]. Thus, we propose that physical activity has a significant effect on sleep quality (Hypothesis 1).

## 1.2 The mediating role of self-control

The self-control theory and related empirical findings suggest that college students' self-control may be an important mediating variable of the relationship between physical activity and sleep quality [36]. Self-control refers to an individual's ability to suppress immediate impulses and regulate his/her behavior to conform to social norms and long-term goals [37]. A strength model demonstrates that self-control, akin to muscle strength, can be enhanced through appropriate exercises, including weightlifting, resistance training, and aerobic activities [26]. Studies in the field of cognitive neuroscience have shown that mild exercise induces cortical activation in the dorsolateral left prefrontal and frontal pole regions and that the activation of these brain regions is associated with self-control increase [38]. Overall, physical activity is significantly positively correlated with both self-control and sleep quality in college students; furthermore, self-control can partially mediate the relationship between physical activity and sleep quality [39]. Individuals with diminished self-regulatory capabilities are frequently prone to yielding to immediate temptations in the vicinity of bedtime, which often results in excessive smartphone use and subsequent delays in bedtime. This behavior, in turn, contributes to suboptimal sleep quality [40]. Physical activity is an exercise pathway that can enhance people's self-control. Taekwondo training positively affects self-control, self-efficacy, and school-life satisfaction [27]. Studies have also shown that the relationship between self-control and sleep is complex and bidirectional and that there is a significant relationship between physical activity level and self-control. On the one hand, the adoption of a healthy lifestyle, which encompasses regular physical activity, sufficient sleep [41], constructive habits [42], effective emotional regulation [43], and proficient stress management techniques [44], all significantly contribute to the enhancement of an individual's self-control. Regular engagement in chronic

physical exercise positively influences executive function, which underpins the capacity for self-regulation [45]. On the other hand, metacognitive control over intrusive and unwanted thoughts at bedtime significantly influences sleep quality [46]. Individuals with weak self-control are more likely to have their bedtime delayed by other distractions, thus reducing their sleep duration and resulting in sleep problems such as sleep deprivation and fatigue [47]. Many scholars believe that poor sleep quality and insomnia are closely related to pre-bedtime cognitive arousal (i.e., thought control) [48, 49]. Hence, we propose that self-control mediates the relationship between physical activity and sleep quality (Hypothesis 2).

## 1.3 The mediating role of mobile phone addiction

Mobile phone addiction, also known as "mobile phone dependence" or problematic mobile phone use, is a non-substance addiction. Notably, the non-substance addiction (or behavioral addiction) emcompasses pathological gambling, food addiction, internet addiction, and mobile phone addiction, whereas the substance addiction (or drug addiction) is a neuropsychiatric disorder characterized by a recurring desire to continue taking the drug despite harmful consequences [50]. Mobile phone addiction may be the biggest non-drug addiction in the 21st century (Richard, 2012). Due to the development of Internet technology and the impact of the COVID-19 pandemic, mobile phone addiction among Chinese college students has been on the rise in the past decade (2013–2022) [51]. Scholars' attention on mobile phone addiction has shifted away from merely describing this phenomenon towards understanding its detrimental effects while exploring preventive measures [52, 53]. Individuals with excessive cell phone usage exhibited significantly reduced daily step counts and reported a higher incidence of neck pain and ocular discomfort. Furthermore, such individuals were more prone to engage in nocturnal cell phone use, which can disrupt circadian rhythms and potentially result in dysregulation of melatonin production, thereby contributing to the onset of emotional disorders, including depression [54]. During adolescence, cell phone abuse can interfere with healthy activities and habits, especially the duration and quality of sleep. For instance, a Finnish study found that that longer mobile phone use was associated with shorter sleep duration [55]. An Iranian study indicates a significant influential relationship between excessive mobile phone use and general health and sleep quality [56]. A study in India revealed a significant positive correlation between participants' smartphone addiction and insomnia severity [57]. A recent study conducted among Chinese adolescents revealed that the prevalence rates of cell phone addiction and poor sleep quality were 26.2% and 23.1%, respectively. Moreover, the incidence of poor sleep quality was found to be significantly higher within the group exhibiting cell phone addiction [58]. Extensive research has demonstrated that poor sleep quality has emerged as a prominent consequence of excessive cell phone use [59–61]. Additionally, some studies show that mobile phone dependence is correlated with physical activity. For example, physical exercise was significantly negatively correlated with mobile phone addiction among college student [61]. Zhong et al. also showed that physical activity has a negative effect on college students' mobile phone dependence [36]. Physical exercise has been shown to ameliorate the psychological and mental symptoms associated with cell phone addiction within a study group by modulating the functions of both the endocrine and immune systems [62]. Additionally, engagement in physical activity can enhance the cognitive function of the hippocampus, potentially serving as a mechanism to mitigate mobile phone addiction [63]. The Health Action Process Approach model intervention significantly reduced the level of mobile phone addiction among adolescents [64]. Accordingly, we propose that mobile phone addiction mediates the relationship between physical activity and sleep quality (Hypothesis 3).

## 1.4 Chain mediation of self-control and mobile phone addiction

Self-control may play a major linking role in mobile phone addiction, i.e., lower self-control suggests increased social media use, whereas individuals with high self-control are more adept at controlling social media use [24]. One meta-analysis revealed that smartphone addiction in adolescents was strongly negatively correlated with self-control [25]. The inability to control oneself is one of the main causes of all addictive behaviors [65]. Social support [66], emotion regulation [43], and stress management [44] all play a role in self-control. Personal factors, including self-motivation [67] and availability of leisure time, alongside environmental factors such as familial support [68] and climatic conditions, are primary determinants affecting an individual's engagement in physical activity. Low self-control is often associated with excessive mobile phone use through interpersonal and transactional modeling [69]. In addition, scholars have verified that physical activity variables are negatively correlated with mobile phone dependence variables and positively correlated with self-control variables. Depression, anxiety, alterations in brain connectivity, and genetic predispositions may all serve as risk factors for mobile phone addiction [54]. They also found that self-control variables are negatively correlated with mobile phone dependence and that self-control partially mediates the effect of physical activity on mobile phone dependence, with the mediating effect accounting for 39.68% of the total effect [70]. Excessive smartphone use may mediate the relationship between physical inactivity and sleep quality issues among students, suggesting that high levels of smartphone engagement, in conjunction with sedentary behaviors, could exacerbate sleep-related problems [71]. In summary, self-control is highly correlated with cell phone addiction. Moreover, both self-control and cell phone addiction appear to mediate the relationship between physical activity and sleep quality. Therefore, we propose that self-control and mobile phone addiction chain-mediate the correlations between physical activity and sleep quality (Hypothesis 4).

## 2. Research methodology

### 2.1 Procedure and participants

Participants were recruited from October 8 to October 30, 2023, from different regions of China (Northeast, North, Northwest, East, Central, South, and Southwest). The studies involving human participants were reviewed and approved by the Ethics Committee of Chengdu Sport University (〔2023〕150) on October 8, 2023. The study was conducted in accordance with the principles of the Declaration of Helsinki, as revised in 2013. The participants were informed that partaking in the study was voluntary and that completing the questionnaire via an online platform indicated their informed consent to voluntarily participate in the study. The participants were to read the research descriptions and consent form and agree to participate, and the survey was designed to end automatically if they did not agree. Once the questionnaire has been submitted, the author has permanent rights to the data. The questionnaire included the purpose and method of research, anonymity and confidentiality regarding participation, and the choice to agree or withdraw from participation, participant requirements (1. enrolled students, 2. no physical disabilities, 3. no mental illness), as well as the confirmation that the survey will not be used for purposes other than research. To mitigate response bias in the online questionnaire, several measures were implemented. Respondents were restricted to one submission per IP address to prevent multiple entries. Additionally, we employed a one-question-per-page format to enhance focus and reduce cognitive load. We assured respondents of the anonymity and confidentiality of their responses to minimize social desirability bias. A total of 2274 undergraduates from 34 universities were surveyed via the online questionnaire (https://www.wenjuan.com/s/UZBZJv98V2/).The participants were 18–24 years old

**Table 1. Descriptive data for main variables.**

|  | Number |
|---|---|
| Male, n% | 743(32.7%) |
| Female, n% | 1531(67.3%) |
| Grade |  |
| Freshman | 28(1.2%) |
| Sophomore | 259(11.4%) |
| Junior | 1349(59.3%) |
| Senior | 638(28.1%) |
| Age in years, M(SD) | 19.18(1.02) |
| Physical activity, n(%) |  |
| Low | 1584(69.66%) |
| Moderate | 327(14.38%) |
| High | 363(15.96%) |
| Sleep quality, M(SD) | 8.61(3.59) |
| Self-control, M(SD) | 2.67(0.61) |
| Mobile phone addition, M(SD) | 3.15(0.93) |

(M = 19.18, SD = 1.02). Females accounted for 67.3% of participants. 45.1% of the subjects used smartphones for over five years. More details were shown in Table 1.

## 2.2 Research tools

**2.2.1 Physical activity scale.** Physical activity was assessed using the Physical Activity Rank Scale-3 [72]. The scale assesses physical activity levels in terms of intensity, frequency, and duration, employing a 5-point Likert scale. The exercise amount is calculated using the formula 'Exercise amount = Exercise frequency * (Exercise duration in minutes—1) * Exercise intensity,' where the highest possible score is 100 and the lowest possible score is 0, the higher the score, the higher the physical activity level. This scale has been widely used in China in recent decades due to the high reliability and validity [73]. The Cronbach's alpha coefficient for the questionnaire in this study is 0.823.

**2.2.2 Sleep quality scale.** The Chinese version of the Pittsburgh Sleep Quality Index (PSQI) was used to assess subjective sleep quality[74]. The PSQI consists of 7 sections: sleep quality, time to sleep, sleep duration, sleep efficiency, sleep disorders, hypnotic drugs, and daytime dysfunction. Each section is scored from 0 and 3. The final score of the PSQI is obtained by cumulatively adding up the scores of each section, and the higher the score, the worse the sleep quality. The Cronbach's alpha coefficient for the scale is 0.911. The reliability and validity of this scale have been reported in the Chinese college student population [75].

**2.2.3 Self-Control Scale.** The Chinese version of Tangney's Self-Control Scale [76] was adopted herein. The scale consists of five dimensions, Resisting Temptation, Healthy Habits, Abstaining from Entertainment, Impulse Control, and Focused Work, and is based on a 5-point Likert scale. The scale includes 19 questions. A higher total score indicates better self-control. The scale has shown high reliability and validity among Chinese college students [77]. The Cronbach's alpha coefficient of the scale is 0.713.

**2.2.4 Mobile phone addiction scale.** The degree of mobile phone addiction among college students was assessed using the Smartphone addiction scale-short version (SAS-SV) [78]. The scale consists of 10 questions on a 6-point Likert scale (1: "Strongly Disagree", 6: "Strongly Agree") and assesses smartphone addiction by self-report. The higher the score, the more

severe the mobile phone dependence. The scale shows high reliability and validity in Chinese college students [79]. The Cronbach's alpha coefficient for this scale was 0.885.

## 2.3 Research procedures

The data analysis tool is SPSS 26.0 software, which mainly carries out descriptive statistics, confidence tests, common method bias tests, correlation analysis, hypothesis tests, etc. Descriptive statistics are used to understand the basic distribution pattern of the data initially. Confidence tests and common method bias tests are used to ensure that the quality of the questionnaire that meets the requirements of data analysis. The correlation analysis constitutes the premise and basis of regression analysis. Based on the above test results, the regression analysis method is used to infer the statistics of the original hypothesis. In addition, the study employed chi-square tests (post-hoc analyses) to investigate variations in poor sleep quality (PSQI score $\geq$ 8) and mobile phone addiction detection rates (SAS-SV score $\geq$ 32) among university students across different genders and academic years. The significance test result of $p < 0.01$ will be used as the critical criterion for the establishment of the hypothesis. In addition, to minimize the error for the test of the mediating effect in the model, the bootstrap method will be used to sample the data 5,000 times to correct any deviations.

## 3. Research results

### 3.1 Commonly used method deviation tests

Given that the data are exclusively derived from questionnaires, there is a potential issue of common method bias arising from the uniform and directional systematic errors in the data results, due to the consistent data collection method. Therefore, it is essential to employ appropriate analytical methods for testing. Typically, Harman's one-way factor analysis is used to assess common method bias by including all Likert-scale questions in the questionnaire. In this study, SPSS 26.0 was utilized to conduct principal component factor analysis on the data results. The findings revealed that the first-factor interpretation rate of the scale was 24.096%, significantly lower than threshold of 40%, indicating that common method bias was not prominent and our data is applicable to further analysis.

### 3.2 Descriptive statistics and correlation analysis for each variable

Correlation analysis is a statistical analysis method used to study the correlations between two or more random variables of equal importance, serving as a prerequisite and foundation for regression analysis. The correlation coefficient, typically represented by "r" ranging from -1 to 1, indicates the strength and direction of the correlation. A negative correlation coefficient suggests an inverse relationship between variables, while a positive correlation coefficient signifies a direct relationship. In this study, Pearson correlation analysis was employed to assess the associations among physical activity, self-control, sleep quality, and mobile phone addiction.

 The correlation coefficient matrix (**Table 2**) reveals significant correlations among the variables. First, physical activity was significantly correlated with sleep quality (r = -0.541, $p < 0.001$), i.e., a higher value of physical activity is indicative of a lower sleep quality index (i.e., a higher sleep quality), consistent with Hypothesis 1. Second, physical activity was significantly correlated with self-control (r = 0.234, $p < 0.001$) and mobile phone addiction (r = -0.367, $p < 0.001$), respectively. Third, sleep quality was also significantly correlated with self-control (r = -0.540, $p < 0.001$) and mobile phone addiction (r = 0.572, $p < 0.001$), respectively. This correlation analysis suggests that both self-control and mobile phone addiction affect the correlations between physical activity and sleep quality. Notably, there is a positive correlation

**Table 2. Descriptive statistics and correlation analysis.**

| | M | SD | Physical activity | Self-control | Mobile phone addiction | Sleep quality |
|---|---|---|---|---|---|---|
| Physical activity | 18.80 | 21.20 | 1 | | | |
| Self-control | 63.28 | 11.57 | 0.234*** | 1 | | |
| Mobile phone addiction | 31.50 | 9.32 | -0.367*** | -0.414*** | 1 | |
| Sleep quality | 8.61 | 3.59 | -0.541*** | -0.540*** | 0.572*** | 1 |

***p<0.001

**p<0.01

*p<0.05

between sleep quality and mobile phone addiction, i.e., the higher the mobile phone addiction, the higher the sleep quality index score and the worse the sleep quality. These results satisfy the prerequisites for regression analysis.

### 3.3 Relationship between physical activity and sleep quality: A chain mediation model

Based on the above correlation analysis, we constructed a chain mediation effect model (Fig 1) with physical activity as the independent variable, sleep quality as the dependent variable, and self-control and mobile phone addiction as mediator variables. The stepwise regression analysis was employed to examine the influence of physical activity on sleep quality. Additionally, we assessed the presence of a significant chain mediation effect of self-control and mobile phone addiction in this relationship. Furthermore, we explored the bidirectional associations between sleep quality and the variables of self-control, cell phone addiction, and physical activity using regression analyses.

In the variable regression relationship test, the covariance VIF of physical activity, self-control, and mobile phone addiction variables were 1.167, 1.218, and 1.330, respectively. All of these values were below 2, suggesting that there were no serious covariance problems between the variables. Physical activity had a significant positive regression effect on self-control (standardized regression coefficient $\beta = 0.234$, $p < 0.001$), a significant negative regression effect on mobile phone addiction ($\beta = -0.286$, $p < 0.001$), and a significant negative regression effect on sleep quality ($\beta = -0.351$, $p < 0.001$). Self-control showed a significant negative regression effect on mobile phone addiction ($\beta = -0.347$, $p<0.001$), and a significant negative regression effect on sleep quality ($\beta = -0.331$, $p < 0.001$). Mobile phone addiction had a significant positive regression effect on sleep quality ($\beta = 0.306$, $p < 0.001$) (**Table 3**). Therefore, the

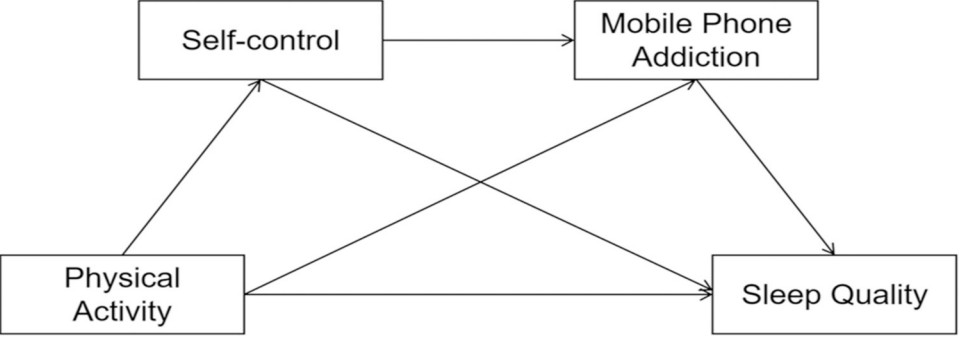

**Fig 1. The chain mediation effect model.**

**Table 3. Regression coefficient test.**

| Outcome variable | Predictor variable | R | R2 | F | β | t |
|---|---|---|---|---|---|---|
| Self-control | Constants | 0.234 | 0.055 | 132.171*** | 2.796*** | 168.471 |
| | physical activity | | | | 0.234*** | 11.497 |
| Mobile phone addiction | Constants | 0.498 | 0.248 | 375.137*** | 1.969*** | 23.655 |
| | physical activity | | | | -0.286*** | -15.268 |
| | self-control | | | | -0.347*** | -18.528 |
| Sleep quality | Constants | 0.738 | 0.544 | 903.234*** | 0.800** | 2.871 |
| | physical activity | | | | -0.351*** | -22.916 |
| | self-control | | | | -0.331*** | -21.188 |
| | mobile phone addiction | | | | 0.306*** | 18.744 |

***p < 0.001

**p < 0.01

*p < 0.05

relationship between the regression effects of each variable in the model was verified. There was also a bidirectional effect between sleep quality and self-control, as well as mobile phone addiction. Analyses indicated that sleep quality exhibited a significant negative regression effect on mobile phone addiction ($β = -0.541$, $p < 0.001$), and conversely, a significant positive regression effect on mobile phone addiction ($β = 0.572$, $p < 0.001$). Post-hoc analyses revealed that female students($χ^2 = 42.68$, P<0.001) and those in their fourth year($χ^2 = 23.80$, P<0.001)of college were more prone to poor sleep quality, and that the level of mobile phone dependence was statistically higher among female students compared to male students($χ^2 = 14.26$, P<0.001). Regression analysis indicated that physical activity had a significant effect on sleep quality, with an eta-squared value of 0.293, and that mobile phone addiction similarly affected sleep quality, with an eta-squared value of 0.327. These findings suggest that both physical activity and mobile phone addiction are significant factors influencing sleep quality. Additionally, self-control exerted a significant influence on mobile phone addiction (with an eta-squared value of 0.171) and sleep quality (with an eta-squared value of 0.292), although the effect sizes were slightly smaller compared to physical activity and mobile phone addiction. The study's findings imply that increases in physical activity and self-control may potentially contribute to enhancements in sleep quality and reductions in mobile phone addiction.

The process function was used to calculate the amount of each mediating effect in the model. To minimize the error, each parameter was sampled 5000 times using the bootstrap method for bias correction. According to the calculation results (**Table 4**), the total effect of

**Table 4. Bootstrap test for mediating effects.**

| | Effect | BOOT SE | BOOT LLCI | BOOT ULCI | Relative mediation effect |
|---|---|---|---|---|---|
| Total effect | -0.541 | 0.018 | -0.576 | -0.506 | |
| Direct effect | -0.351 | 0.015 | -0.381 | -0.321 | |
| Total indirect effect | -0.190 | 0.011 | -0.212 | -0.168 | 35.16% |
| Indirect effect 1 (self-control) | -0.078 | 0.007 | -0.091 | -0.063 | 14.36% |
| Indirect effect 2 (mobile phone addiction) | -0.088 | 0.007 | -0.101 | -0.075 | 16.18% |
| Indirect effect 3 (self-control & mobile phone addiction) | -0.025 | 0.003 | -0.031 | -0.019 | 4.60% |

Indirect effect 1: physical activity → self-control → sleep quality

Indirect effect 2: physical activity → mobile phone addiction → sleep quality

Indirect effect 3: physical activity → self-control → mobile phone addiction → sleep quality.

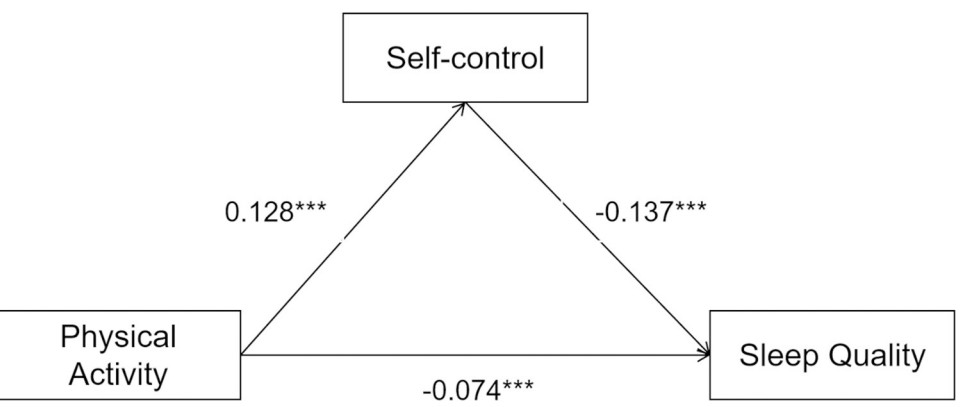

**Fig 2. Model of mediating roles of self-control between physical activity and sleep quality.**

physical activity on sleep quality in the model is –0.541. The total indirect effect amount is –0.190, accounting for 35.16% of the total effect, with the 99% confidence interval of [–0.212, –0.168]. There were three indirect effects: self-control, mobile phone addiction, and self-control & mobile phone addition. They accounted for 14.36%, 16.18%, and 4.60% of the total effects, respectively. Their confidence interval (99%) is [–0.212, –0.168], [–0.091, –0.063], and [–0.101, –0.075], respectively. The specific paths are presented in Figs 2–4.

## 4. Discussion

First, this study shows that physical activity levels were negatively correlated with college students' sleep quality index. The PSQI scale scoring rules indicate that the higher the PSQI score, the poorer the quality of sleep; thus, the higher the physical activity level, the better the sleep quality (i.e., the lower the frequency of sleep disorders). This study validated Hypothesis 1, consistent with previous findings [39, 80]. A meta-analysis showed that scientific and reasonable physical activity can improve sleep quality [81]. Students with high levels of physical activity have longer nighttime sleep, fewer awakenings, and better sleep quality [82]. A cross-sectional study also shows that college students who are physically active enough also have better sleep quality [12, 83]. This implies that participation in various types of physical activities effectively promotes sleep quality and reduces sleep disorders. However, some studies have

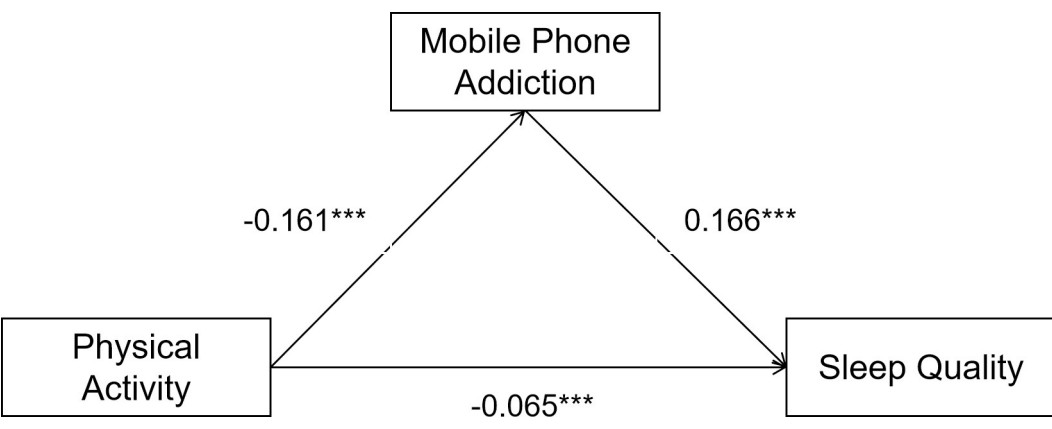

**Fig 3. Model of mediating roles of mobile phone addiction between physical activity and sleep quality.**

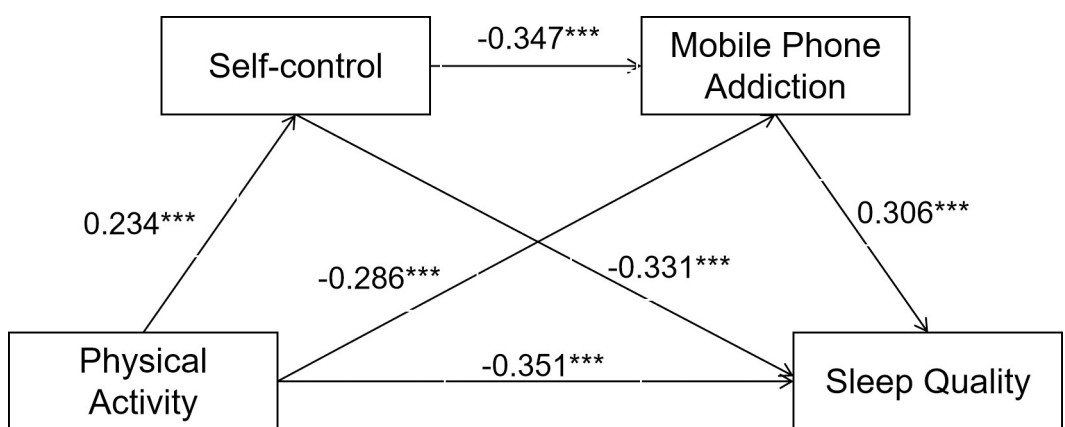

**Fig 4. Model of chain mediating roles of self-control and mobile phone addiction between physical activity and sleep quality.**

also shown that physical activity does not have a significant effect on sleep quality in young people [84], and that high-intensity physical activity at night has no significant effect on sleep quality [85]. This investigation utilized a large-scale cross-sectional study to examine the relationship between physical activity and sleep quality within a university student population, contributing to the growing body of evidence suggesting that physical activity exerts a significant influence on sleep quality in young adults. Majd A. Alnawwar's review indicates that engaging in physical activity during the morning or afternoon may confer greater benefits to sleep quality compared to intense exercise performed at night or in the evening. The study suggests that moderate-intensity aerobic exercises, strength training regimens, as well as mind-body practices like yoga and tai chi, can enhance the quality of sleep. Moreover, the research finds an association between improved sleep quality and a frequency of four to seven weekly sessions of aerobic exercise, coupled with three sessions of moderate-intensity exercise per week [35]. The intensity of physical activity adopted and the duration of physical activity performed to enhance sleep quality warrant further in-depth exploration in future. Moreover, Chinese college students face unique social pressures and cultural expectations that may affect their sleep patterns and mobile phone habits. The emphasis on academic excellence in Chinese society often results in extended study periods, potentially contributing to sleep deprivation. Additionally, Confucian values, which underscore social harmony, could influence students' self-regulation and perception of mobile phone addiction. Further investigation into these cultural subtleties is essential to understand their implications for the psychological and behavioral health of college students worldwide.

Second, this study showed that self-control mediates the relationship between physical activity and sleep quality, and thus Hypothesis 2 was validated. Physical activity significantly and positively predicted self-control, while self-control negatively predicted sleep quality, i.e., the higher the level of physical activity, the greater the self-control, the lower the PSQI score, and the better the sleep quality. This is consistent with previous findings [86]. An intervention study has found that sleep-related self-regulation techniques can reduce poor sleep habits and enhance the quality of sleep [87]. A meta-analysis showed modest positive associations between self-control and physical activity, healthier eating and healthier sleep [88]. Correlational meta-analytic studies have shown that acute exercise at multiple intensities can enhance individuals' executive functioning [89]. Cognitive neuroscience studies have also determined that mild exercise induces cortical activation in the dorsolateral left prefrontal and frontal pole regions, and that the activation of these brain regions is associated with increased self-control

[38]. This better explains the role of self-control as a mediator between physical activity and sleep quality. However, it has also been proposed that better quality and quantity of sleep is influenced by factors outside of self-control, such as environment [90] and individual differences [91]. One theoretical framework posits that self-control is a finite internal resource, which, when depleted, impairs one's ability to process information effectively and to regulate cognition, behavior, and emotion. An alternative model suggests that motivation is a primary determinant of self-control failure. Consequently, a variety of factors, including stress, self-control beliefs, motivation, social support, familial influences, and temperament, all contribute to the dynamics of self-control capacity [66].

Third, this study suggests that mobile phone addiction has a mediating effect on the correlations between physical activity and sleep quality. As the level of physical activity increases, the mobile phone dependence index and sleep quality index decrease, and sleep quality improves. The results of this study consolidate Hypothesis 3 and agree with previous findings [71]. According to immersion theory, when college students are immersed in mobile phones, their endocrine systems are in a state of continuous activity, gradually releasing adrenaline, dopamine, and other excitatory hormones that keep the body in a state of excitement, leading to disruption in life patterns, which in turn affects the quality of sleep [92, 93]. In addition, mobile phone addiction causes college students to spend a significant amount of time playing games on their mobile phones, which reduces the amount of time spent in sleeping and exercising, which in turn affects the quality of their sleep [94]. Therefore, controlling the time spent on mobile phones to reduce mobile phone addiction and increase opportunities for physical activity among college students can help improve the quality of sleep among college students. While some studies have shown that physical activity can reduce mobile phone use, others have pointed out that in some cases, physical activity may increase reliance on mobile phones, especially when individuals use them to record exercise data or share exercise achievements [95]. In conjunction with physical activity levels, lifestyle factors such as smoking and dietary habits are recognized as pivotal risk factors affecting sleep quality. Moreover, psychological well-being, including perceived stress and anxiety, exerts a profound influence on the quality of sleep. Additionally, physical factors such as chronic pain and fatigue significantly contribute to the degradation of sleep quality [12].

Finally, this study indicates that self-control and mobile phone addiction are closely related. They constitute intermediate links in the pathway: physical activity → self-control → mobile phone addiction → sleep quality influence, and there is a chain-mediated effect in the process of physical activity influencing sleep quality. Specifically, college students who score higher on physical activity tend to have lower PSQI scores, indicative of better sleep quality, due to their greater self-control and reduced phone addiction. A strong correlation is observed between self-control and mobile phone addiction, i.e., higher levels of self-control are associated with lower mobile phone dependence and vice versa. However, the effect sizes of chained mediators were not as high as those of mediators alone, probably due to the negative correlation between self-control and mobile phone addiction, and thus their interaction may weaken the effect of chained mediators when both are used together as mediating variables. This finding supports Hypothesis 4 and aligns with previous studies [96, 97]. The present study confirms that self-control and mobile phone addiction have chain-mediated effects on the process of physical activity, affecting college students' sleep quality. In addition, another study highlights the role of anxiety and depression as a chain mediator in the relationship between physical activity and sleep quality [98]. It has also been shown that addiction to short videos directly affects the sleep quality of college students, indirectly through physical activity and procrastination behaviors [99]. Thus, complex interactions exist between these variables, necessitating further investigation.

Overall, physical activity has a positive effect on promoting college students' levels of self-control and improving mobile phone dependence. Through self-control, mobile phone dependence can be effectively contained, thereby improving sleep quality. At the same time, improving physical activity levels also helps reduce college students' mobile phone dependence. Therefore, the chain mediation model of physical activity → self-control → mobile phone addiction → sleep quality proposed in this study is feasible. Based on the findings of our study, we recommend that universities incorporate courses addressing the scientific use of mobile phones and sleep health education into their curricula. Additionally, we suggest that universities encourage students to engage more actively in sports clubs and extracurricular activities, as these measures may enhance sleep quality among university students.

This study, however, is not without its limitations. First, our study was cross-sectional, and its predictions failed to reveal the causal relationships between physical activity and sleep quality. Second, the questionnaire we used to measure college students' physical activity level, self-control level, mobile phone dependence index, and sleep quality was subjective. In addition, this study did not account for other variables that might influence sleep quality, such as mental health issues including anxiety and depression, as well as environmental factors like lighting and noise. However, the conclusions reached and the proposed intervention strategies in this study may be applicable to university students in other countries, or they may be more specifically relevant to Chinese university students, depending on the nature of the corresponding research. Future research endeavors should incorporate longitudinal studies to elucidate the temporal effects of physical activity on sleep quality. Additionally, the establishment of intervention trials designed to examine the mediating roles of variables such as self-regulatory capacity and cellular phone dependency will provide further insights into the nuanced relationship between physical activity and sleep quality. For instance, intervention trials could be designed to examine the impact of various physical activities, such as long-distance running and rock climbing, on both mobile phone addiction and sleep quality. Consequently, a promising direction for future research involves utilizing pedometers to obtain more objective measures of physical activity, as well as employing actigraphs for more precise assessments of sleep quality. Such methodologies will enable a more scientific and rational elucidation of the relationship between physical activity and sleep quality.

## 5. Conclusion

In this study, we explored the mechanism underlying how physical activity influences college students' sleep quality through a cross-sectional survey study. We proposed a chain mediation model: physical activity → self-control → mobile phone addiction → sleep quality, which provides ideas for college students to improve their sleep quality. The results of this study offer valuable insights and serve as a reference for future investigations into the mechanisms through which physical activity influences the enhancement of sleep quality in college students.

## Supporting information

**S1 Data.**
(XLSX)

## Author Contributions

**Conceptualization:** Zhonggen Yin, Chengbo Yang.

**Data curation:** Tong Liu.

**Formal analysis:** Tong Liu.

**Funding acquisition:** Yanhong Zhang.

**Investigation:** Jialiang Yu, Shuting Huang.

**Software:** Xiaomei Yu.

**Writing – original draft:** Zhonggen Yin.

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
