## [Decision Letter · Decision Letter 0]

8 Oct 2024

PONE-D-24-34938The relationship between physical activity and sleep quality among college students: The chain-mediating effects of self-control and mobile phone addictionPLOS ONE Dear Dr. Yang,

Thank you for submitting your manuscript to PLOS ONE. After careful consideration, we feel that it has merit but does not fully meet PLOS ONE’s publication criteria as it currently stands. Therefore, we invite you to submit a revised version of the manuscript that addresses the points raised during the review process.

We look forward to receiving your revised manuscript.

Kind regards,

Mohammad Sidiq, PhD Pain Sciences Physiotherapy

Academic Editor

PLOS ONE

Journal Requirements:

**Additional Editor Comments:**

Dear Authors, I have some suggestions and recommendations for your paper as follows

1. Introduction:

   Explain why the chain-mediation model is critical to the discussion early in the article. Discuss the significance of examining self-control as well as the mobile phone addiction as the mediators, specifically in the light of contemporary students’ life.

   - Broaden the global relevance: In a larger part, the introduction is dedicated to Chinese students, or rather their experiences and perceptions of living with the respondents. To widen the scope, it is only suitable to briefly explain how the conclusions given might apply to college students in other areas.

 2. Methods:

   – Improve the understanding of the measures of the main constructs such as self-control, physical activity, and mobile phone addiction. Make it easier and shorter for the readers especially when it comes to the particular scales used in the research.

   It is about time to explain the limitations of self-reported data more elaborately. Although this is done, more detailed consideration of response bias especially in an online survey could have been done in a more detailed manner thus enhancing the credibility of the methodology.

   - Consider inclusion of objective measures: One of the future directions for the study is to employ more objective measures for physical activity using pedometers or more objectively measure sleep quality using actigraphs.

 3. Results:

   - First, it will give more detail of the specific nature of the practical significance of the findings. First, while the significance and nature of the correlations, regression coefficients are well reported, how big these differences are in real world terms (ie, effect sizes) would be an addition.

   - Add more visual representation: While tables are given, a graphic display to explain the chain mediation model (apart from the flowchart) might be helpful to elucidate the directions of the variables to the audience.

4. Discussion:

   - Address conflicting evidence: Subsequently you cite some papers showing that physical activity has no overall impact on the quality of sleep. Adding on those two aspects would provide a better reason for considering why your study differs, thus enhancing the representation of the work’s credibility.

   - Elaborate on cultural context: Because this study concerns Chinese college students, it might have been more useful to pay more attention to potential relaters that could explain sleep, mobile phone addiction, or self-control in some way in this particular group, especially in the contrast with the effect surveyed in Western or other culture.

   - Explore potential intervention strategies: Nevertheless, such recommendations or ideas for the improvement of sleep quality as an intervention (e.g., decreasing the use of a mobile phone through special university programs) would be more valuable.

5. Limitations and Future Research:

   - Expand the limitations section: If you want to qualify your work you pointed out the cross-sectional design as a weakness, it will better to talk about other variables which could have an influence but were not included in your study for example mental health issues like anxiety or depression.

   - Suggest more detailed future research directions: In discussing future work you say that you still require experimental studies or longitudinal studies etc., it will be more helpful if you were to give some possible set up of such studies like how some reduction in the use of mobile phones can be tested experimentally through randomized controlled trials.

   - Potential for generalization: Briefly, it is possible to state that the conclusions obtained can be applied to college students in other counties as well as only to the Chinese students depending on the type of the correspondent study.

Reviewers' comments:

Reviewer's Responses to Questions

**Comments to the Author**

1. Is the manuscript technically sound, and do the data support the conclusions?

Reviewer #1: Yes

2. Has the statistical analysis been performed appropriately and rigorously? 

Reviewer #1: No

3. Have the authors made all data underlying the findings in their manuscript fully available?

Reviewer #1: Yes

4. Is the manuscript presented in an intelligible fashion and written in standard English?

Reviewer #1: Yes

5. Review Comments to the Author

Reviewer #1: Dear Author

Kindly mention figure no. 1 with the relevant text, where the chain mediating effect is highlighted. Also, include post hoc analysis and suggest to do bidirectional statistical analysis.

Regards

6. PLOS authors have the option to publish the peer review history of their article (what does this mean?). If published, this will include your full peer review and any attached files.

Reviewer #1: **Yes: **JYOTI SHARMA

---

## [Author Response · Author response to Decision Letter 0]

18 Nov 2024

Dear Reviewer:

Thank you for your valuable comments on this study, which provided us with ideas for further analyses of the relationship between physical activity and sleep quality, including post-hoc analyses, bivariate analyses, and so on. Based on the full discussion and understanding of your comments, we spent a great deal of time on revision, and we sincerely hope that the revised manuscript will meet your expectations. Thank you again for your comments and suggestions.

---

## [Editor Report · Decision Letter 1]

4 Dec 2024

The relationship between physical activity and sleep quality among college students: The chain-mediating effects of self-control and mobile phone addiction

PONE-D-24-34938R1

Dear Dr. Yang

We’re pleased to inform you that your manuscript has been judged scientifically suitable for publication and will be formally accepted for publication once it meets all outstanding technical requirements.

Kind regards,

Mohammad Sidiq, PhD Pain Sciences Physiotherapy

Academic Editor

PLOS ONE

Additional Editor Comments (optional):

Dear Chengbo Yang, thank you for addressing the reviewers comments positively. I am satisfied with the revision and i suggest that your article be accepted for publication.
---

## [Editor Report · Acceptance letter]

9 Dec 2024

PONE-D-24-34938R1 

PLOS ONE

Dear Dr. Yang, 

I'm pleased to inform you that your manuscript has been deemed suitable for publication in PLOS ONE. Congratulations! Your manuscript is now being handed over to our production team.

Kind regards, 

on behalf of

Dr. Mohammad Sidiq 

Academic Editor

PLOS ONE